# Large-magnitude events unlikely in induced earthquake sequences

Linxuan Li ⬤ ✉, Kyungjae Im ⬤ & Jean-Philippe Avouac ⬤

Injecting fluids into the subsurface frequently triggers earthquakes, yet the maximum size of these events remains difficult to forecast. Earthquake frequencies are typically assumed to decay exponentially with magnitude, a model that fits natural seismicity well. Here, we use a global compilation of injection-induced sequences to show that this model breaks down for about half of the cases, which instead show a downturn in the occurrence of larger events. Numerical simulations and observed spatial patterns of induced seismicity suggest that heterogeneous loading near injection sites and disorganized fault networks may limit the growth of large ruptures. These results provide a possible explanation for why induced earthquakes often remain below magnitude 2–3 and provide a basis for evaluating seismic hazard before and during injection operations.

Earthquakes induced by human activities pose a growing societal concern as they can reach magnitudes exceeding 5[1–11]. The risk of such events, which are often related to subsurface fluid injection, can hinder geological carbon storage and geothermal energy production at the scales required to substantially reduce $CO_2$ emission[12]. In recent years, substantial progress has been made in understanding the governing physical mechanisms and in the development of forecasting models[13–18]. These models are commonly calibrated using the rates of small-magnitude events (typically $M < 2$) and can, in principle, be extended to evaluate the probability of larger, potentially damaging earthquakes when coupled with a magnitude–frequency distribution (MFD) model[19] that characterizes the ratio of small to large events.

The most widely used MFD is the Gutenberg–Richter (GR) model (Methods, Eq. 1; blue in the inset of Fig. 1a)[20], which assumes an exponential decay in earthquake frequency with magnitude, characterized by the $b$-value. The GR model has proven robust across tectonic settings and scales ranging from laboratory to global environments[21–24]. The physical origin of the GR distribution remains uncertain. It could stem, for example, from the structural self-organization of a fault system[25], or from dynamic interactions among earthquakes[26].

However, the applicability of the GR model to induced seismicity is debated. Physics-based models, whether conceptual or derived from fracture mechanics, indicate that the maximum magnitude ($m_{max}$) of induced events is constrained by the injected fluid volume, the perturbed rock volume, or stratigraphic barriers, unless tectonic stress is fully mobilized[27–31]. In such cases, a tapered GR (TGR) distribution

(Methods, Eq. 2; red in the inset of Fig. 1a)[32], which further decreases the probability of large earthquakes near and above the corner magnitude ($m_{corner}$), or a truncated GR distribution[33], which imposes a hard upper magnitude cutoff, may be more appropriate. Yet, such upper limits might not be observable. Some studies argue that induced earthquakes conform to the TGR model[34,35], whereas others find no departure from the standard GR distribution[36,37], and still others report both behaviors[38].

In this study, we test whether and why induced earthquakes are better described by the GR or TGR distribution, using a global high-quality dataset drawn from more than 100 inject'ion-induced seismicity cases worldwide, together with advanced statistical analyses[39] and physics-based numerical simulations[40]. We also develop an approach to identify MFD types and to forecast maximum magnitudes in real time during injection operations. We focus on the TGR model because it provides a statistically tractable alternative to the truncated GR model, whose hard upper limit typically cannot be resolved within the duration of most earthquake catalogs[41]. Even if such a hard limit were to exist, the TGR model remains an appropriate approximation for finite earthquake catalogs (Supplementary Fig. 1), with the added benefit of being conservative.

## Results

### Observed magnitude–frequency patterns

We selected 38 injection-induced seismicity sequences with data quality sufficient for detailed analysis of the MFD out of over 100

Division of Geological and Planetary Sciences, California Institute of Technology, Pasadena, CA, USA. ✉e-mail: lxli@caltech.edu

sequences across six continents, associated with hydraulic fracturing, enhanced geothermal systems, underground gas storage, waste fluid disposal, conventional geothermal reservoirs, and small-scale scientific experiments (Supplementary Data 1 and 2). Our dataset incorporates data from previous studies[28,36,37,42–52] and approximately 20 additional recent cases. Our selection criteria are: (1) sequences must be well-recorded from the beginning of the operations, avoiding cases where seismic stations were only deployed after the largest event; (2) there must be a clear temporal and spatial correlation between seismicity and industrial operations; (3) magnitudes must be reported on a consistent scale; and (4) each sequence must include at least 100 events above the magnitude of completeness. Based on these criteria, we retain 38 sequences for further analysis (Supplementary Data 2).

To assess whether the observed MFDs deviate from the GR model and are better described by the TGR model, we apply the likelihood-ratio test, the most effective for distinguishing the two models[39] (see "Methods" for details). The test inherently accounts for the difference in model complexity between the GR and TGR distributions. If the returned $p$-value ($p_{LRT}$) is small (e.g., <0.05 or 0.1), we reject the GR model in favor of the TGR model.

In approximately half of the 38 sequences (18 using a $p_{LRT}$ threshold of 0.05 and 22 using 0.10), the GR model can be confidently rejected, with the TGR model providing a superior fit (Fig. 1a and Supplementary Data 2). In sequences with low $p_{LRT}$, visual inspection of the MFDs also reveals pronounced deviations from the GR distribution: the observed distributions fall outside the 90% confidence interval of the best-fitting GR model, whereas the TGR distribution closely matches the observations (insets in Fig. 1b and Supplementary Fig. 2). The remaining half of the cases, which exhibit relatively high $p_{LRT}$ values, either follow a GR distribution or a TGR distribution whose tail has not yet been sampled. In some cases, earthquake sequences have been terminated following the occurrence of a relatively large event because operational activities were halted. Such termination breaks the stationarity of the sequence and can cause the largest observed events to appear near the end of the catalog[36]. To assess whether the observed TGR behavior could arise from sequences that intrinsically follow a GR distribution but are prematurely terminated after the occurrence of a relatively large event, we performed synthetic tests. The results show that this type of termination does not produce TGR characteristics (Supplementary Fig. 3).

The MFD model enables forecasts of the expected maximum magnitude based on the catalog size and the estimated distribution parameters (see Methods). For sequences with low $p_{LRT}$ values, the GR model systematically overestimates the maximum magnitude (Fig. 1c), whereas the TGR model more accurately reproduces the observations and the fact that the magnitude rarely exceeds M2–3 in many sites (Fig. 1d). Excluding small-scale experiments, which have a maximum magnitude inherently limited by their scale, the estimated $m_{corner}$ of the TGR cases is predominantly between 0 and 3 (especially 1–2) (Fig. 1e). The absence of larger corner magnitudes in the TGR cases is not due to a detection limit (Supplementary Fig. 4), but rather reflects a characteristic range of $m_{corner}$ for induced seismicity sequences exhibiting tapered MFDs. Compared to the GR model, the TGR model predicts a much slower increase in the expected maximum magnitude as event counts grow, once the tapered tail is sampled (Supplementary Fig. 5). This suggests that at most sites where the current maximum observed magnitude is below M3, continued operations are unlikely to substantially elevate the seismic hazard.

For the larger sequences, we further assess whether they consistently follow either the GR or TGR model by analyzing the temporal evolution of $p_{LRT}$ (see "Methods"). While some sequences persistently conform to one model, others do not, with TGR characteristics emerging at the beginning, middle, or end of a sequence (Supplementary Fig. 6). No systematic pattern is observed in the temporal evolution of MFD parameters (Supplementary Fig. 7).

## Correlations with other characteristics

We examine whether the GR and TGR sequences differ systematically across a range of characteristics (see Methods; Supplementary Data 2; Fig. 2, Supplementary Figs. 8–10): catalog features (catalog size, magnitude type, MFD parameters, seismicity duration, seismicity rate, earthquakes per injected volume, inter-event time variability, and the spatial extent, fractal dimension, and migration of seismic clouds), operational variables (injection type, total injected volume, peak flow rate, and peak wellhead pressure), and tectonic context (historical seismicity, stress regime, earthquake depth, and lithology). To identify potential bias arising from scale differences in some parameters, we compare GR and TGR sequences using both (1) the full dataset and (2) only field-scale operations, excluding small-scale experiments. For cases with transitions between GR and TGR behavior, we also assess whether the shifts coincide with changes in operational or statistical properties.

Relative to the GR sequences, the TGR sequences are characterized by shallower focal depths and smaller seismic cloud volumes (Fig. 2a–c). However, we do not observe a systematic difference between sedimentary formations and crystalline basement cases (Supplementary Data 2). No correlation is observed between the number of events and the seismic cloud volume (Supplementary Fig. 10), suggesting that larger volumes reflect broader regions influenced by fluid injection rather than simply larger catalogs. In both cases, the rupture radii of the largest induced earthquakes are much smaller than the seismic cloud size (Fig. 2d), indicating that the maximum magnitude is not constrained by the size of the perturbed volume. In addition, TGR sequences tend to exhibit less oblate (i.e., more spherical) seismic clouds (Fig. 2e) and higher fractal dimensions of pairwise earthquake distances (Fig. 2f). These observations indicate that TGR sequences typically occur within more complex three-dimensional fault networks, whereas GR sequences tend to outline more planar structures resembling a well-oriented fault plane. This behavior is consistent with the intuitive notion that simpler, structurally more mature geometries favor larger ruptures. It is consistent with Schultz et al. [38]'s suggestion that the activation of a finite fracture system may favor bounded earthquake magnitude growth, while the reactivation of pre-existing faults may favor unbounded ruptures. The statistics shown in Fig. 2 are calculated using only field-scale operations, given the potential scale dependence of some parameters (Fig. 2a–c). For parameters not expected to depend on scale (Fig. 2d–f), analyses including small-scale experiments yield the same patterns. Apart from these geometric characteristics, the GR and TGR cases exhibit similar properties, except that GR cases tend to have larger maximum magnitudes.

## Physical controls

To investigate the mechanisms that may cause tapering of the MFD, we conduct three-dimensional simulations of earthquake sequences on a planar fault perturbed by constant fluid injection (see Methods). The fault follows laboratory-derived rate-and-state friction laws[53]. We explore two end-member cases: a homogeneous fault (Fig. 3a) and a fault with self-affine heterogeneity (Fig. 3e). For reference, we also simulate a tectonic earthquake sequence driven solely by constant shear loading. Our simulations continue until seismicity ceases, when shear stress is fully released.

Under constant tectonic loading, the homogeneous fault produces periodic -M4.7 events. For injection-induced sequences, building on previous results that initial stress governs rupture behavior[8,30,54–58], we test two initial shear stress levels: 4 MPa below and 2 MPa above the estimated dynamic shear strength (i.e., the shear stress at rupture termination) referred to as understressed and overstressed conditions[55], respectively. In the understressed regime, the ruptures remain confined within the perturbed region, expanding gradually until they reach the fault boundaries (Fig. 3b). In contrast, overstressed conditions permit

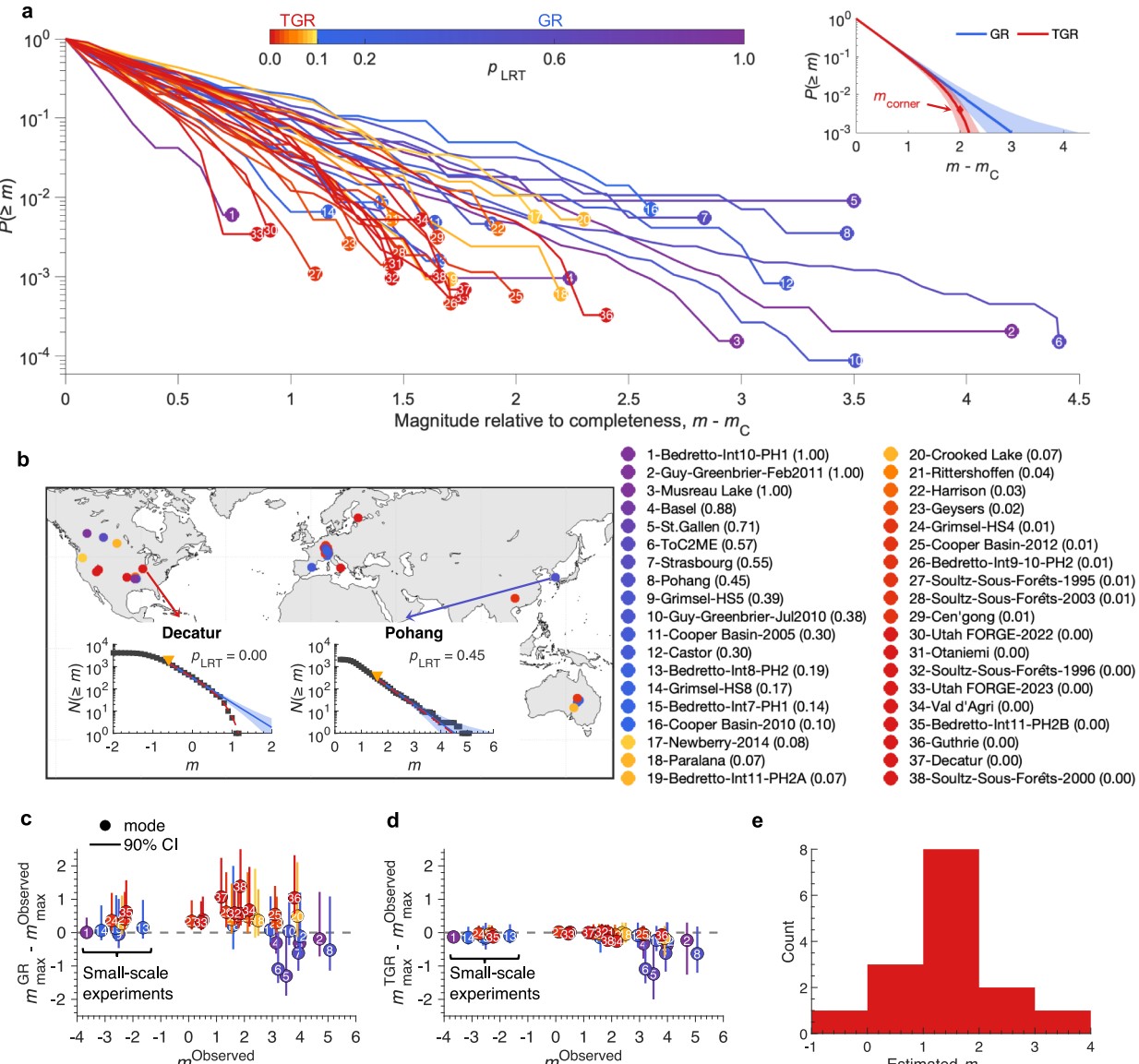

**Fig. 1 | Magnitude–frequency distributions of induced seismicity in a global compilation reveal departures from the Gutenberg–Richter distribution.**
**a** Observed magnitude–frequency distributions (MFDs) of 38 cases, shown as the frequency of earthquakes with magnitudes greater than or equal to a given value. The x-axis is shifted by each case's magnitude of completeness ($m_C$), aligning all curves at zero. Colors represent the likelihood-ratio test p-value ($p_{LRT}$). Circles mark the largest observed event in each case, labeled by case index sorted in descending order of $p_{LRT}$. Values in parentheses in the legend denote the $p_{LRT}$ for each case. The inset shows example theoretical Gutenberg–Richter (GR; blue) and tapered Gutenberg–Richter (TGR; red) distributions with their 90% confidence intervals (shaded areas). **b** Geographic distribution of the 38 cases, colored by $p_{LRT}$. Two examples, Pohang (GR) and Decatur (TGR), are highlighted. Black squares indicate observations, yellow triangles mark $m_C$, blue lines show the best-fitting GR models with 90% confidence intervals (shaded), and red curves denote the best-fitting TGR models. **c** Difference between the predicted maximum magnitudes under the GR model and the observed values, showing systematic overestimation for TGR-type cases. Circles indicate modal predictions; vertical bars represent 90% confidence intervals accounting for sampling and parameter uncertainty. **d** Same as **c**, but using TGR predictions. **e** The distribution of estimated corner magnitudes for cases with $p_{LRT} \leq 0.05$, excluding small-scale experiments.

ruptures beyond the injection zone, limited only by the fault extent (Fig. 3c). When the full earthquake sequence is considered, both stress regimes yield MFDs with a relative excess of smaller events and slightly lower maximum magnitudes compared with the tectonic reference (Fig. 3d). This behavior arises because the injection imposes heterogeneous loading, promoting rupture nucleation even where spontaneous failure would not occur otherwise, impeding rupture growth into stronger regions. Elevated pore pressure further suppresses large events by reducing the stress drop relative to tectonic earthquakes and by promoting aseismic creep[59], which releases the shear stress. These features are robust across a range of injection rates and healing times (Supplementary Fig. 11).

When injection into the heterogenous fault begins early or late in an interseismic interval (Fig. 3f, g), sequences respectively analogous to the understressed and overstressed cases are observed. Simulations of far-field injection, represented as a uniform reduction in effective normal stress, produce MFDs that more closely resemble those from tectonic loading (dashed curves in Fig. 3g), a pattern also observed for the homogeneous fault. In nature, similar responses are also expected when loading is modulated by poroelastic stresses induced by distant fluid sources[60], in addition to direct pore-pressure changes. The difference resulting from near- and far-field injection is consistent with the observation that GR sequences often exhibit larger, deeper, and more diffuse seismic clouds (Fig. 2a–c).

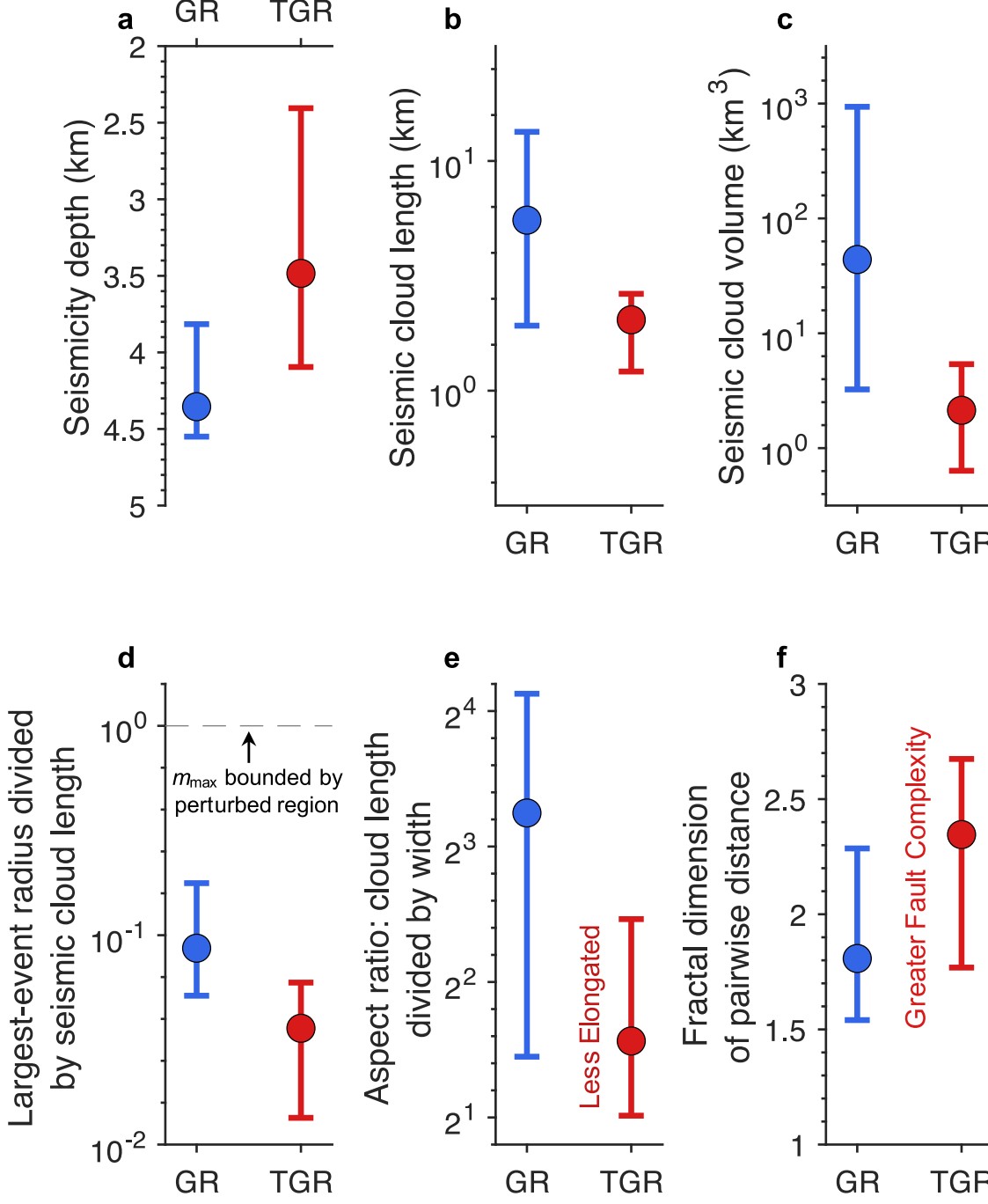

**Fig. 2 | Tapered Gutenberg–Richter behavior is linked to smaller fluid-affected regions and greater fault complexity.** The earthquake catalog is classified into Gutenberg–Richter (GR) and tapered Gutenberg–Richter (TGR) cases using a likelihood-ratio test with a $p_{LRT}$ threshold 0.05. Bars represent first and third quartile bounds and circles indicate median values. **a** Median depth of the earthquake swarms. **b** Seismic cloud length, defined as the semi-major axis of the smallest ellipsoid enclosing ≥ 90% of seismicity. **c** Seismic cloud volume, defined by the volume of this ellipsoid. **d** Estimated rupture radius of the largest observed event divided by the ellipsoid's semi-major axis. Rupture radii are computed assuming a circular-crack model with a stress drop of 1 MPa, representing an upper bound of the radii. The largest earthquakes are markedly smaller than the inferred fluid-perturbed region. **e** Aspect ratio of the seismic cloud, defined as the ratio of the semi-major to the semi-minor axis. **f** Fractal dimension of pairwise event distances, where larger values indicate more complex fault geometries. All quantities are shown for field-scale operations only; small-scale experiments are excluded. Values for each case, as well as the full cumulative distribution functions, are provided in Supplementary Figs. 8 and 9, respectively.

Collectively, these heterogeneous fault simulations reproduce the spectrum of behaviors observed in natural systems (except for the variability of b-value because the fractal properties are not changed). In all cases, the finite fault size ultimately constrains the upper magnitude and yields comparable MFDs across stress regimes, though the timing of large events varies with the initial stress state. The dimensions of the largest events in the real cases remain much smaller than those of the fluid-influenced zone (Fig. 2d), suggesting that they are similarly limited by the sizes of faults within the perturbed volume rather than by the extent of the perturbation itself. Moreover, the more spherical geometry and higher fractal dimension of the TGR sequences (Fig. 2e, f) imply an additional structural control not captured by our

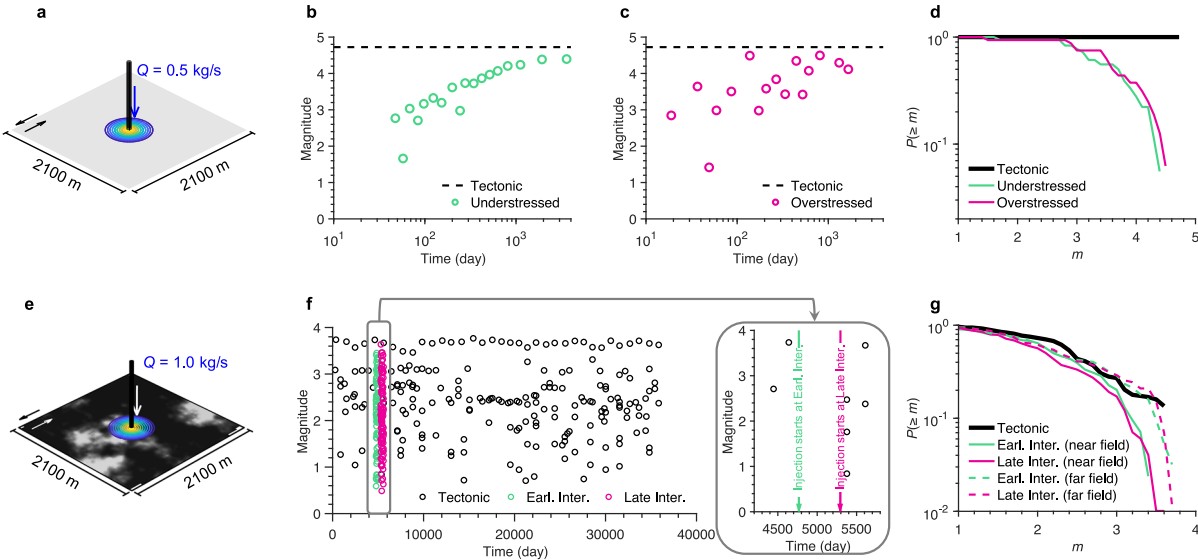

**Fig. 3 | Simulated earthquake sequences reveal that near-field injection promotes smaller events due to heterogeneous loading.** Numerical simulations are performed on two-dimensional rate-and-state faults embedded in an elastic half-space. **a–d Homogeneous fault. a** Model setup: a uniform fault with constant frictional properties is subjected to fluid injection at its center (representing near-field injection). Fluid diffuses radially within a porous reservoir of finite thickness parallel to the fault. Simulations are conducted under two initial shear stress conditions: an understressed case, where the initial shear stress ($\tau_{ini}$) is 4 MPa lower than the nominal dynamic shear stress at rupture arrest ($\tau_{dyn}$), and an overstressed case, where $\tau_{ini}$ exceeds $\tau_{dyn}$ by 2 MPa. **b, c** Magnitude–time distributions. Green and magenta denote the injection-induced sequences, and the dashed line marks the characteristic event size under tectonic loading. **d** Magnitude–frequency distributions (MFD) of the simulated catalogs. The thick black line indicates tectonic

sequences composed of repeating characteristic events (drop-off beyond the characteristic magnitude not shown). **e–g** Heterogeneous fault. **e** Model setup showing spatial variability in the rate-and-state parameter contrast ($a_{RS} - b_{RS}$); gray and black areas denote velocity-weakening and velocity-neutral patches, respectively. The heterogeneous normal-stress field is shown in Supplementary Fig. 15. **f** Magnitude–time distributions for tectonic and injection-induced (near-field) sequences. Injection simulations use initial conditions extracted from two stages of the tectonic sequence (see inset): early interseismic (day 4767) and late interseismic (day 5295). **g** MFDs for tectonic, near-field, and far-field injection cases. Far-field injection (dashed curves) is simulated by a uniform normal-stress reduction of 1 MPa/yr from the same initial conditions as the near-field cases. For tectonic sequences, MFDs are plotted up to 0.1 magnitude units below the largest event to avoid finite-size effects. See Supplementary Movies 1–5 for rupture evolution.

single-fault simulations. Injection-induced earthquakes can activate fault networks that do not have to be compatible with active tectonic loading. Such reactivated faults could have formed in different tectonic episodes, need not be mechanically or geometrically coherent, producing a more disorganized and spatially fragmented rupture pattern.

Regardless of the underlying origin of GR statistics, near-fault fluid injection can shift the MFD toward an overabundance of smaller earthquakes compared with tectonic sequences. When the pre-existing distribution of seismogenic patches already limits large ruptures, the superposition of injection effects may further reinforce the predominance of small events. As operations proceed, new fault systems might be activated and more faults will be under far-field conditions, temporal transitions between MFDs may occur.

**Monitoring seismic hazard during operations**

Based on our results, we propose a real-time strategy for managing seismic risk during injection operations. The strategy shares the same objective as previously proposed traffic-light systems in the literature[12,38]. The GR model should be assumed initially. The $b$-value estimated from the initial seismic sequence, and operational scenarios can be used[15–18], to define a maximum allowable injected volume for a specified acceptable magnitude. As operations proceed, the recorded seismicity can be used to assess whether the GR model remains valid or whether a TGR distribution provides a better fit. This assessment can be performed in real time using $p_{LRT}$ derived from expanding and sliding windows (see Supplementary Text 1). Using MFD parameters estimated from the current data, the expected maximum magnitude and allowable injection volume can be forecast.

We use two contrasting case studies, Decatur and Basel, to demonstrate the strategy's performance (Fig. 4). Decatur, Illinois launched the first US demonstration-scale carbon capture and storage project in 2011. Basel, Switzerland hosted the first modern enhanced geothermal system project, halted after a damaging M3.4 earthquake in 2006. In Decatur, both expanding and sliding windows show consistently low $p_{LRT}$ from the early stages (Fig. 4a), in strong support of the TGR model. In contrast, in Basel, $p_{LRT}$ from expanding window remains low until the occurrence of the largest event, whereas sliding window yields relatively high $p_{LRT}$ values during most of the time before the largest event (Fig. 4b). Because expanding windows over the same interval length show low $p_{LRT}$ values, this discrepancy suggests a temporal transition: the Basel sequence, initially TGR, evolved toward GR before the mainshock (see Supplementary Text 1). Predictions of the maximum magnitude align closely with the observations (Fig. 4c, d). In contrast, in Decatur, assuming a GR model would significantly overestimate the maximum magnitude (Supplementary Fig. 12a), whereas applying a TGR model in Basel fails to forecast the largest event in advance (Supplementary Fig. 12b).

Effective application of this framework requires dense local seismic monitoring networks. Because most observed TGR cases have corner magnitudes between 0 and 3 (Fig. 1e), the magnitude of completeness must be at least several units smaller, which is challenging to achieve with regional networks. Nevertheless, knowledge of the typical corner-magnitude range can provide valuable early warnings and help avert highly damaging events. A notable example is the Pohang enhanced geothermal system project in South Korea, which is thought to have caused the most damaging earthquake (2017 Mw 5.5) to strike the Korean Peninsula in centuries[3]. Only about 200 earthquakes spatially associated with the drill site were recorded prior to the

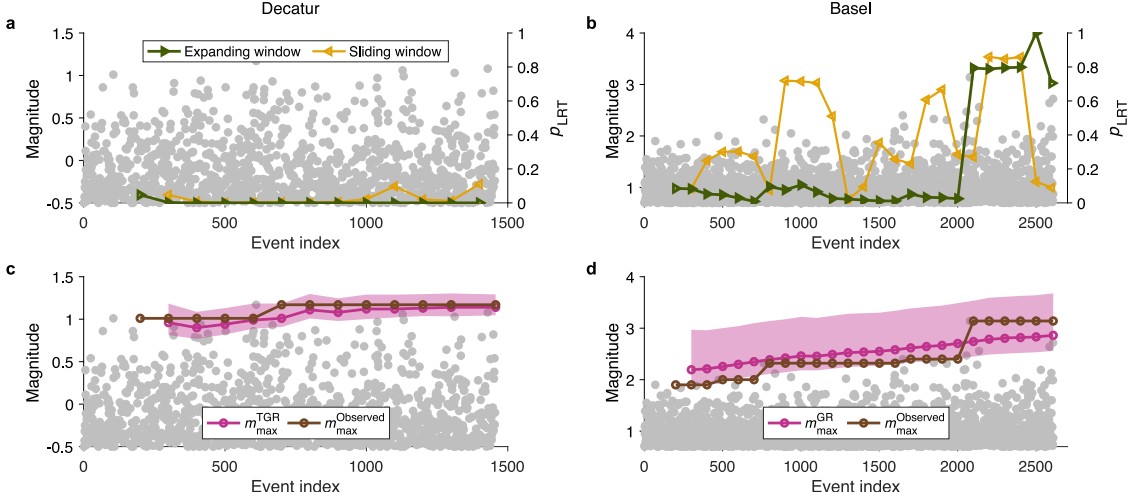

**Fig. 4 | Determining the appropriate magnitude–frequency distribution model allows real-time seismic risk monitoring.** Temporal evolution of $p_{LRT}$ for Decatur (**a**) and Basel (**b**), calculated using only events preceding each time step. Results are shown for expanding (green) and sliding (yellow) windows. The initial expanding window includes 200 events, with each subsequent window adding 100 events; the sliding window comprises the 200 temporally nearest events. Values are assigned to the last event in each window. Decatur consistently follows the tapered Gutenberg–Richter (TGR) model, whereas Basel transitions between the tapered Gutenberg–Richter and Gutenberg–Richter (GR) models. Observed and predicted maximum magnitudes as functions of time for Decatur (**c**) and Basel (**d**). Observed values are shown in brown, and predicted maximum magnitudes in purple, with shaded areas representing the 90% confidence interval from random sampling. The same forward-expanding windows as in **a** and **b** are used. Because predictions depend on magnitude–frequency distribution parameters estimated from the preceding window, the first window shows only observed maximum magnitudes. The TGR model was applied to Decatur and the GR model to Basel; predictions using the alternative models are shown in Supplementary Fig. 12.

mainshock[61]. Fitting a GR model yields a magnitude of completeness near 0.5, above which only about 50 events remain, with a $b$-value of approximately 0.7. This limited catalog prevents robust likelihood-ratio testing. However, notably, a Mw 3.2 earthquake occurred seven months before the mainshock, during a stimulation[61]. If the underlying distribution were TGR with a corner magnitude smaller than 3, the probability of such an event would have been extremely low (Supplementary Fig. 5). Thus, in practice, observing an induced event of this size warrants a preference for a GR model and implies a high seismic risk.

This study demonstrates that induced seismicity is often less hazardous than commonly assumed. Near-fault fluid injection may suppress the occurrence of large earthquakes by imposing heterogeneous loading. The local distribution of seismogenic patches, shaped by fault geometry, material properties, and stress heterogeneity, may further modulate the magnitude–frequency behavior. Together, these findings highlight the interplay between fluid processes and fault mechanics in controlling induced earthquake statistics. The proposed real-time monitoring framework enables continuous evaluation of evolving seismic hazard during injection operations and provides a practical basis for safer and more adaptive subsurface resource management.

## Methods
### Determine the MFD and temporal variations
In the classic unbounded GR distribution[20], the probability of an earthquake with a magnitude greater than or equal to $m$ is:

$$P(\geq m) = 10^{-b(m_C - m)}, \quad m \geq m_C \quad (1)$$

where $m_C$ is the magnitude of completeness, and the $b$-value controls the relative frequency of large and small magnitudes. Another parameter is the so-called $a$-value, which is equal to the base-10 logarithm of the number of events with magnitude no less than 0.

The TGR distribution incorporates an additional decay at the tail of the GR distribution[32,62]. The TGR distribution can be expressed in terms of magnitude as:

$$P(\geq m) = 10^{-b(m_C - m)} \exp\left[10^{\frac{3}{2}(m_C - m_{corner})} - 10^{\frac{3}{2}(m - m_{corner})}\right], \quad m \geq m_C \quad (2)$$

where $m_{corner}$ is corner magnitude, which controls where the deviation from the GR distribution occurs.

To estimate the MFD, we first determine $m_C$, which remains a topic of active research. Following Li and Avouac[39], we take a conservative approach by selecting $m_C$ as the maximum of three estimates: (1) literature-reported values; (2) the maximum curvature method with a + 0.2 magnitude correction[63]; and (3) the $b$-value stability method[63,64]. We also use the plot of magnitude versus event index for events above the selected $m_C$ to ensure there is no significant detection gap caused by temporal variations in the magnitude of completeness, for example, due to changes in the seismic network or short-term aftershock incompleteness.

We exclude all events below $m_C$ and apply the maximum likelihood estimation (MLE) to fit both GR and TGR models[32,65]. Parameter uncertainties are quantified using the bootstrap method, drawing resampled catalogs of equal size with replacement. The confidence interval of the MFD can be obtained using the semi-analytical solution proposed by Li and Avouac[39].

To determine whether a catalog better fits a GR or TGR distribution, we use the likelihood-ratio test[32], shown by Li and Avouac[39] to be the most robust discriminator. This test compares the maximum log-likelihoods of the GR and TGR models and returns a $p$-value ($p_{LRT}$). A low $p$-value (e.g., <0.05 or 0.1) supports rejecting the GR model in favor of the TGR model. The likelihood-ratio test inherently accounts for the difference in model complexity between the two distributions. The statistic $2LR$, defined as twice the difference in maximum log-likelihood between the models, follows a chi-square distribution whose degrees of freedom correspond to the difference in the number of model parameters. As a result, the more complex TGR model is only favored when it provides a sufficiently better fit to the data. Synthetic tests have shown that the likelihood-ratio test rarely misclassifies GR distributions as TGR (false positive rates are near the defined $p$-value threshold), though TGR distributions may be misclassified as GR when

their tails are undersampled[39]. If a threshold of 0.05 is selected, the identification accuracy exceeds 50% when $a/b - m_{corner} > 0.5$, and exceeds 90% when $a/b - m_{corner} > 1$[39]. To further validate low $p_{LRT}$ values, we compute the empirical probability of obtaining lower or higher $p_{LRT}$ under the optimal GR and TGR models using a large ensemble of synthetic catalogs, mitigating the risk of false significance due to multiple testing. The random catalogs following a given GR distribution are generated using the inversion method[66], while those following a TGR distribution are generated using the method proposed by Vere-Jones et al. [62]. We find that such a high frequency of low $p_{LRT}$ values we observe is unlikely to arise by chance if all the underlying distributions were truly GR.

Among the 38 spatial samples analyzed, 19 contain more than 600 events above $m_C$, while the remaining 19 have fewer than 500 events. For the larger catalogs, we assess the temporal evolution of the $p_{LRT}$ using a method introduced by Li and Avouac[39]. The catalog with a size of $N_{total}$ is partitioned into $WI$ non-overlapping subcatalogs, each containing between $N_{min} = 100$ and $N_{max} = 1000$ events. We fit GR and TGR models to each subcatalog, compute the corresponding $p_{LRT}$, and assign this value to all events within the subcatalog. This process is repeated 200 times for each $WI$, and the median $p_{LRT}$ across runs is retained. We vary $WI$ from $\lceil N_{total}/N_{max} \rceil + 1$ to $\min\{\lfloor N_{total}/N_{min} \rfloor - 1, \lceil N_{total}/N_{max} \rceil + 3\}$ to capture robust first-order trends. Meanwhile, the temporal evolution of the MFD parameters is obtained. While this approach cannot eliminate all random fluctuations, synthetic tests confirm its ability to recover the dominant temporal patterns[39]. We define cases in which the $p_{LRT}$ never falls below 0.05 as exhibiting a persistent GR distribution. Conversely, cases where the $p_{LRT}$ never exceeds 0.1 are considered to exhibit a persistent TGR distribution. Cases with $p_{LRT}$ that fall between these thresholds are treated as potential transition cases. These are examined manually by segmenting the catalog according to the temporal evolution of the $p_{LRT}$. We assess whether subcatalogs associated with high $p_{LRT}$ (greater than 0.1) significantly deviate from the TGR distribution inferred for the remaining periods or the entire catalog, given the null hypothesis of no transition. An example is shown in Supplementary Fig. 13. In cases where high $p_{LRT}$ values appear attributable to random fluctuations, we classify them as "Persistent TGR". The remaining cases, in which a clear and sustained shift is observed between GR and TGR distributions, are classified as transition cases.

## Forecasting of maximum magnitude

Given a known catalog size $n$ (i.e., the number of earthquakes with magnitude no less than $m_C$), we can estimate the expected distribution of maximum magnitude due to random sampling under a specified MFD using both empirical and theoretical approaches.

The empirical approach simulates numerous synthetic catalogs following the defined MFD, to extract the distribution of maximum magnitudes. From this, we obtain the mode and the 5th and 95th percentiles.

The theoretical approach derives the cumulative distribution function (CDF) of the maximum magnitude by raising the CDF of the MFD to the $n$-th power[36]. For the GR model, the mode of the maximum magnitude is[36]:

$$\widehat{m_{max}} = m_C + \frac{\log_{10} n}{b} = \frac{a}{b}, \tag{3}$$

For large $n$ (as considered in this study), the confidence interval bounds are given by[36]:

$$\widehat{m_q} = m_{max} - \frac{1}{b}\log_{10}\left[-\ln q\right], \tag{4}$$

where $\hat{m}_q$ is the magnitude corresponding to a given quantile $q$. In this regime, the separation between the modal value and the percentiles becomes effectively independent of the catalog size. The difference between this approximation and the exact expression derived by van der Elst et al. [36] scales as $\frac{1}{2b}\frac{\ln q}{\ln 10}\frac{1}{n}$. For the 5th and 95th percentiles, this difference is smaller than 0.01 magnitude units when $n \geq 100$ and $b \approx 1$, which is typical for the sequences analyzed here. Therefore, the approximation in Eq. (4) is adequate for the catalog sizes considered in this study. For the TGR model, no analytical solution exists. For practical convenience, we therefore provide empirical functions by fitting the results of numerical simulations (Supplementary Fig. 5). For simplicity, we assume $m_C = 0$. The mode and the 5th and 95th percentiles of the maximum magnitude predicted by the TGR model, as a function of catalog size $n$, can be approximated using a logistic function, with the parameters depending on the $a$- and $b$-values and $m_{corner}$, and are estimated via nonlinear regression on synthetic data.

For each of the 38 catalogs, we compute the mode and 90% confidence interval of the maximum magnitude under both GR and TGR distributions. We account for (1) random sampling uncertainty and (2) parameter estimation uncertainty by generating MFD parameters using both the best-fit model and bootstrapped catalogs. The second source of uncertainty is relatively small, as the MFD parameter estimation uncertainties are minimal in our cases due to the large catalog sizes.

For the 19 catalogs with sufficient catalog size, we compute time-evolving forecasts of the maximum magnitude using an expanding-window approach. Each window begins with the first earthquake and grows by 100 events per iteration. Two strategies are employed: (1) using MFD parameters from the full catalog, and (2) estimating parameters within each window. Parametric uncertainties are not incorporated due to the computational demand. Both methods yield consistent trends. We show the mode evolution obtained using the first strategy in Supplementary Fig. 6. Results based on the 5th and 95th percentiles using the same strategy, as well as those derived from the second strategy, yield qualitatively consistent outcomes.

## Comparison of additional characteristics

To assess whether systematic differences exist between seismicity catalogs better described by GR or TGR models, we analyze a suite of catalog-level, operational, and geological parameters.

Catalog properties include catalog size, magnitude type (local magnitude, moment magnitude, etc.), duration, MFD parameters, seismicity rate (number of $M \geq 0$ earthquakes per day inferred from the GR model), and the seismogenic index[67] ($\log_{10}$ of the total number of $M \geq 0$ earthquakes divided by injected volume). We also compute the global and local coefficients of variation of interevent times ($\Delta t$) that characterize temporal clustering of earthquakes[68–70]. The global coefficient $CV_{\Delta t}$ is defined as the standard deviation divided by the mean of the interevent times, whereas the local coefficient is defined as $LV_{\Delta t} = \frac{3}{n_{\Delta t}-1}\sum_{i=1}^{n_{\Delta t}-1}\frac{(\Delta t_i - \Delta t_{i+1})^2}{(\Delta t_i + \Delta t_{i+1})^2}$, where $n_{\Delta t}$ is the number of interevent intervals. For both metrics, values close to 0 indicate regular event spacing, values near 1 correspond to a Poissonian process, and larger values indicate temporal clustering. The metric $LV_{\Delta t}$ helps detect local variability in interevent times, for example identifying sequences that are globally clustered but locally regular. We further estimate the spatial fractal dimension of pairwise distances[71,72] from the correlation integral $C(r) = \frac{2 n_{pairs}(d < r)}{n_{EQ}(n_{EQ}-1)}$, where $n_{EQ}$ is the total number of earthquakes and $n_{pairs}(d < r)$ is the number of event pairs with hypocenter distance $d$ less than $r$. The fractal dimension $d_f$ is defined as the slope of the scaling region in a plot of $\log C(r)$ versus $\log r$. Values of $d_f$ close to 1, 2, and 3, correspond to seismicity distributed primarily along a line, within a plane, and throughout a three-dimensional volume, respectively. Additional catalog properties include seismic front migration

(defined in the following paragraphs) and the size and shape of the seismic cloud, quantified by the smallest ellipsoid encompassing 90% of events. Operational parameters include injection type, total injected volume, peak injection rate, and peak wellhead pressure. Geological parameters comprise tectonic stress regime, presence of pre-injection tectonic earthquakes, hypocenter depths (5th, 50th, and 95th percentiles), and whether seismicity occurs in sedimentary or basement rocks. The operational and geological parameters are compiled from the sources listed in Supplementary Data 1.

We summarize the qualitative and quantitative features in Supplementary Data 2 and Supplementary Fig. 8, respectively. For quantitative analysis, we compare the empirical cumulative distribution functions (ECDFs) of the GR and TGR samples. In addition to visual inspection of their differences, we apply the Kolmogorov–Smirnov (KS) and Cramér–von Mises (CvM) tests[73] to assess whether the GR and TGR samples are drawn from the same underlying distribution. The KS and CvM tests quantify the maximum and integrated squared differences between the ECDFs, respectively. Empirical $p$-values ($p_{KS}$ and $p_{CvM}$) are estimated from 1000 Monte Carlo resamplings of the pooled data. Small $p$-values indicate statistically significant differences between the two samples. Given the limited sample size (tens of cases), the statistical power of these tests is modest, and genuine differences may not reach formal significance. To account for potential scale dependence (e.g., in maximum magnitude or spatial extent), we perform the analysis on both the full dataset (38 catalogs) and a subset excluding small-scale experiments (29 catalogs; omitting Bedretto and Grimsel).

Seismic front migration denotes the expansion of the volume occupied by earthquake hypocenters over time and is commonly interpreted as reflecting the propagation of a pore-pressure front or a slow-slip front[8,74]. Migration is typically characterized by the distance between the furthest earthquakes and the injection well. Because the coordinates of injection wells are often unavailable in our dataset, we instead analyze distances relative to the first recorded event above the magnitude of completeness as a function of time. Specifically, we examine the temporal evolution of the envelope containing 90% of the events. We find no consistent correlation between migration behavior and MFD type. For example, in several cases we observe transitions from slow to fast seismic front migration, but these transitions are not systematically associated with changes in MFD type.

For catalogs that exhibit transitions between GR and TGR regimes, we compute Pearson correlations between time series of the $p_{LRT}$ and time series of MFD parameters, seismicity rate, and interevent time variability (Supplementary Fig. 14). We exclude spatial metrics from this analysis due to the prevalence of radial expansion, which may bias results. Injection records are instead reviewed manually to check for a correlation with the transition point of the MFD type, given the lack of available digital datasets in most cases. We do not find any correlation between the injection operations and the MFD types.

## Earthquake sequence simulations

We use the open-source software Quake-DFN[40] to simulate spontaneous seismic and aseismic slip on a two-dimensional planar fault embedded in an infinite, three-dimensional elastic medium. The fault is discretized into rectangular planar elements, and slip evolution is computed using the boundary element method by solving the momentum balance equation for each element:

$$M_i^{lumped} \ddot{\delta}_i = \sum_j K_{ij}\left[\delta_j^{t=0} - \delta_j\right] - f_i\left[\sigma_i^{eff,\,t=0} - \Delta p_i\right] - \frac{G}{2v_S}\dot{\delta}_i + \Delta\tau_i^{tectonic}$$

(5)

where subscripts $i$ and $j$ denote the fault elements, and $\delta$, $\dot{\delta}$, and $\ddot{\delta}$ represent slip, slip velocity, and slip acceleration, respectively. $f$ is the coefficient of friction, $\sigma^{eff,\,t=0}$ is the initial effective normal stress, and

$\Delta p$ is the pore pressure perturbation from fluid injection. $\Delta\tau^{tectonic}$ represents tectonic shear loading. The stiffness matrix $K_{ij}$ defines the quasi-static shear stress change on element $i$ due to slip on element $j$[75]. $G$ is the shear modulus and $v_S$ is the S-wave speed. The formulation is quasi-dynamic, with radiation damping approximated by the velocity-dependent term[76] and inertial effects by lumped mass[77]. Fault friction is governed by the laboratory-derived rate-and-state framework[53]:

$$f_i = f_{ref} + a_{RS}\log\frac{\dot{\delta}_i}{V_{ref}} + b_{RS}\log\frac{V_{ref}\theta_i}{D_{RS}},$$

(6)

$$\frac{d\theta_i}{dt} = 1 - \frac{\dot{\delta}_i\theta_i}{D_{RS}},$$

(7)

where, $\theta$ is the state variable, $a_{RS}$, $b_{RS}$, and $D_{RS}$ are material properties, and $f_{ref}$ is the steady-state friction at reference velocity $V_{ref}$. We adopt the aging law[78] (Eq. 7), widely used in earthquake sequence simulations, to describe the evolution of $\theta$.

When the velocity-weakening region exceeds the nucleation size, dynamic rupture is possible. The nucleation length before injection is estimated as[79]:

$$h_{RA} = \frac{\pi G D_{RS} b_{RS}}{2\sigma^{eff,\,t=0}(b_{RS} - a_{RS})^2}.$$

(8)

For injection-driven cases, reduced effective normal stress increases $h_{RA}$. The cohesive zone length before injection is estimated as[80]:

$$\Lambda_0 = \frac{9\pi}{32}\frac{G D_{RS}}{b_{RS}\sigma^{eff,\,t=0}}.$$

(9)

We perform two sets of simulations with distinct loading mechanisms. First, a constant shear loading is applied uniformly to each fault element, such that $\Delta\tau^{tectonic}$ increases with time, while the effective normal stress remains constant. Second, earthquakes are driven by injection-induced pore pressure perturbations, which are modeled by radial fluid diffusion from a wellbore under constant injection rate $Q$ in a reservoir of thickness $w$. Injection occurs at the fault center, and the pore pressure perturbation can be expressed as[81]:

$$\Delta p(r, t) = \frac{Q\eta}{4\pi\kappa w}E_1\left(\frac{r^2}{4ct}\right),$$

(10)

where $r$ is the distance to the injection point, $\kappa$ is permeability, $c$ is hydraulic diffusivity, $\eta$ is fluid viscosity, and $E_1(x) = \int_x^\infty \frac{e^{-\zeta}}{\zeta}d\zeta$ is the exponential integral.

We model a $2.1 \times 2.1$ km fault, a size comparable to seismically activated regions inferred from seismicity clouds (Fig. 2b). Two cases are considered: (1) a fault with homogeneous properties and initial conditions, and (2) a fault with self-affine heterogeneous properties and heterogeneous initial conditions resulting from the cumulative slip of a prior tectonic sequence. For both cases, we use a shear modulus of 24 GPa, an S-wave velocity of 3 km/s, and a Poisson's ratio of 0.25. The lumped mass is $5 \times 10^5$ kg/m², permeability is $3 \times 10^{16}$ m², hydraulic diffusivity is $10^{-3}$ m²/s, fluid viscosity is $0.4 \times 10^{-3}$ Pa s, and fluid density is $10^3$ kg/m³. The porous reservoir has a thickness of 1 m.

For the homogeneous fault, two initial stress states are examined, representing overstressed and understressed conditions relative to the nominal dynamic shear strength:

$$\tau_{dyn} = \sigma^{eff,\,t=0}f_{dyn} = \sigma^{eff,\,t=0}\left[f_{ref} + (a_{RS} - b_{RS})\log\frac{V_{peak}}{V_{ref}}\right],$$

(11)

where $V_{peak} \approx 1$ m/s. Note that the actual shear stress differs from the nominal value because the effective normal stress is modified by pore pressure changes. Simulations include cases where the initial shear stress ($\tau_{ini}$) is 2 MPa above or 4 MPa below $\tau_{dyn}$. The tectonic sequence is simulated by imposing a constant stressing rate of 0.01 MPa/yr. The rate-and-state friction parameters are $a_{RS} = 0.005$, $b_{RS} = 0.009$, and $D_{RS} = 500$ μm, with a reference friction coefficient of 0.6 at a reference slip rate of $10^{-9}$ m/s. The initial normal stress on the fault is 40 MPa. The initial state variable corresponds to 1 yr in the simulation shown in Fig. 3a–d to 1 Ma in the simulation in Supplementary Fig. 11b. The injection rate is 0.5 kg/s in Fig. 3a–d and 2 kg/s in Supplementary Fig. 11c. Under these conditions, the nucleation length is approximately 530 m and the cohesive zone size is about 60 m. A grid spacing of 25 m is adopted to adequately resolve the rupture process. For injection cases, we neglect tectonic loading because induced seismicity lasts much shorter than the >100 years required to recover the typical 1–10 MPa stress drop[82] of a tectonic earthquake.

For the heterogeneous fault, we randomly generated two independent self-affine distributions of normal stress and shear stress using a code from ref. 83 with an isotropic Hurst exponent of 0.7 (Supplementary Fig. 15). We pose intervals in the random generation $-0.004 \leq a_{RS} - b_{RS} \leq 0$ and $1\,\text{MPa} < \sigma_0 < 19\,\text{MPa}$ with uniform $D_{RS} = 1$ mm. Initially, we conducted a simulation with a tectonic loading rate of 3 mm/year, which results in an approximate recurrence interval of 1000 days for a complete rupture (Fig. 3f). We then selected two specific time points—~140 days (early interseismic) and ~640 days (late interseismic)—following a full rupture that occurred on day 4640 (Fig. 3f). The tectonic loading simulation results for these chosen timings are used as initial parameters for the injection-induced earthquake simulation. The injection rate in the induced earthquake simulation is 1 kg/s. To mimic far-field injection, we perform simulations using the same initial conditions; however, instead of applying a spatially heterogeneous pore-pressure field, we impose a reduction in normal stress of 1 MPa/yr along the fault.

## Data availability
The data sources for each induced seismicity site are provided in Supplementary Data 1.

## Code availability
The scripts used for data analysis are available at https://doi.org/10.22002/c85b2-td293[84].

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

## Acknowledgements
We thank Yihe Huang, Julie Albaric, Stefan Baisch, Louis De Barros, Xiaowei Chen, Simone Cesca, Trenton Cladouhos, Kamel Drif, Paul Friberg, Peter Hennings, Won-Young Kim, Grzegorz Kwiatek, Xinglin Lei, Patricia Martínez-Garzón, Riccardo Minetto, and Miao Zhang for providing or discussing their earthquake catalogs. We have benefited from valuable discussions with Stephen Bourne, Junjie Dong, Nadia Lapusta, and Alexandros Savvaidis. J.P.A. discloses support for the research of this work from the National Science Foundation (Award No. 1822214) through the Industry–University Cooperative Research (IUCR) Center for Geomechanics and Mitigation of Geohazards.

## Author contributions
L.L. and J.P.A. conceived the study. L.L. carried out the data analysis. L.L. and K.I. performed the numerical simulations. L.L. and J.P.A. interpreted the results. All authors contributed to writing the paper.

## Competing interests
The authors declare no competing interests.
