## [Transparent Peer Review file · Nature Communications]

Large-magnitude events unlikely in induced earthquake sequences

Corresponding Author: Mr Linxuan Li

Version 0:

Reviewer comments:

Reviewer #1

(Remarks to the Author)
See the attached PDF.

Reviewer #2

(Remarks to the Author)

This study builds on methods developed in a previously published study (Li and Avouac, BSSA, 2025) to examine the magnitude-frequency distributions of induced earthquake sequences and determine whether they are better fit with a standard Gutenberg-Richter (GR) relationship or a tapered GR, and how this can evolve over time during each sequence. The results suggest that the size of induced earthquakes tends to be more limited than expected by the standard GR relationship, and numerical simulations suggest this is due to heterogeneous loading and fault network conditions. The conclusions seem well supported, the manuscript is generally well-written and easy to follow, and the topic is of great interest to both the scientific community and can have practical applications as well when it comes to injection operation management. I only have a few minor comments and suggestions that are aimed at adding a little more detail and/or clarity at certain points listed below.

First, a general question. Perhaps I missed this in either the main manuscript or the Methods section, but why do figures such as Extended Data Fig. 1-2, and Extended Data Fig. 5 show confidence intervals for the GR results but not the TGR results?

L. 287: How large is “large n”?

L. 312-320: It would be helpful to add a little more detail to this paragraph, especially to clarify what some of the parameters are and how they are measured. For example, how is spatial fractal dimension measured? How is seismic front migration measured? And do the operational and geological parameters all come from the sources that are referred to in Supp. Table 1?

L. 321-331: It would be helpful to state that the results of the quantitative analysis can be found in Extended Data Figure 3.

L. 335-340: Likewise, please state where these Pearson correlation results can be seen.

Figure 2 caption (L. 654): Replace “versus” with “divided by”

Figure 4 caption (L. 691): In the sentence beginning “Decatur consistently follows the Gutenberg-Richter (GR) model...” shouldn’t this be the “tapered Gutenberg-Richter” model?

Supplemental Text 1: The second paragraph is repeated.

Supplemental Fig. 6: Please define all the parameters that are shown in the various panels (as in Extended Data Fig. 3).

Supplemental Fig. 7: Why aren't the different sequences numbered as they are in Supp. Fig. 6? Also again please define what the parameters are that are shown in each panel.

Supplemental Fig. 8: In the caption, it would probably be better to change the sentence beginning "The second to last row displays..." to "The second to last rows display..." (or "The, second, third, and last rows..." as in the following sentence), otherwise it sounds like it's only referring to the one row second from the bottom.

Supplemental Table 1: In addition to WFD, the table also has "WWD" listed (e.g., Guthrie, Guy-Greenbrier). I assume this is the same as WFD (wastewater disposal = waste fluid disposal)?

Supplemental Table 2: Please restate what the different operation abbreviations stand for. Also, how is "highly earthquake-active zone or not" defined?

Reviewer #3

(Remarks to the Author)

In this paper, the authors present an analysis of observed magnitude-frequency distributions (MFD) in injection-induced seismic sequences. While they initially compiled over 100 cases worldwide, only 38 sequences were retained for detailed analysis due to the high quality of their data. Using statistical approaches, the authors evaluate whether each sequence follows the standard Gutenberg-Richter (GR) model or a Tapered Gutenberg-Richter (TGR) distribution. Traditionally, the GR model, which assumes a constant exponential decay in earthquake frequency with magnitude, is used to characterize seismicity. However, the authors observed that the standard model is inadequate in approximately half of the analyzed cases. Instead, a TGR distribution is preferred, as it accounts for a sharp downturn in the occurrence of larger events near a critical value known as the corner magnitude. This finding indicates that, at many sites, the physical conditions of injection effectively limit the potential magnitude of induced events. The authors conducted three-dimensional numerical simulations to investigate the physical mechanisms causing the tapering of the MFD. These simulations were performed on a planar fault, considering both homogeneous and self-affine heterogeneous properties, subjected to constant fluid injection. For these calculations, they utilized laboratory-derived rate-and-state friction laws to model the interaction between fault surfaces. In the simulations, the authors demonstrate that fault geometry and stress heterogeneity within the fluid-perturbed volume are what constrain the maximum size of earthquakes, rather than the spatial extent of the perturbation itself. Finally, the authors propose a practical strategy for real-time seismic risk management during injection operations. The Gutenberg-Richter (GR) model is initially assumed to define the maximum allowable injection volumes based on acceptable magnitude thresholds. As operations proceed, statistical tests, specifically the likelihood-ratio test developed by authors in previous work, are employed to assess whether the seismic sequence continues to follow the GR model or if a Tapered Gutenberg-Richter (TGR) distribution provides a superior fit. This distinction is crucial: a TGR classification indicates that the probability of large-magnitude events is significantly lower than the standard model predicts. This framework allows operators to adjust injection parameters, such as volume and pressure, to mitigate the hazard and prevent the triggering of damaging earthquakes.

The manuscript is clearly written, the methodology is thoroughly explained, the data sources are provided in the Supplementary Table 1, and the code used is properly referenced.

The methodological approach is innovative and compelling. It establishes a fundamental framework for artificial injection systems, providing a robust basis for evaluating seismic hazard before and during subsurface operations.

I recommend that the paper be accepted as it is.

Version 1:

Reviewer comments:

Reviewer #1

(Remarks to the Author)

Review of Nature Communications article MS# NCOMMS-25-099928-T,

The revised manuscript of Li et al., "Large-magnitude events unlikely in induced earthquake sequences" is an article concerning the shape (and cause) of the statistical distribution of induced earthquake magnitudes.

In this revised version, the authors have suitably addressed all of my concerns – I am happy to recommend it for publication now.

There's just one small typo I noticed, after re-reading: on Line 132, "Schutz" should be "Schultz".

Thanks,
-Ryan

A. Response to Reviewer #1 (Ryan Schultz):

The manuscript of Li et al., “Large-magnitude events unlikely in induced earthquake sequences” is an article concerning the shape of the statistical distribution of induced earthquake magnitudes. The authors use statistical methods to identify cases of induced seismicity that deviate from unbound magnitudes. They perform numerical earthquake sequence simulations to suggest that this deviation can be related to heterogeneous loading of stress on a rough fault. Generally, the paper is well-written, logical, timely, and covers an interesting topic. I feel that the results of the paper could be interesting the readership of Nat. Comm. As a caveat, my expertise with earthquake sequence simulations is limited to just the implications.

That said, I feel there are additions/revisions that are needed to improve this paper before I would recommend it for publication. In particular, the manuscript could bolster some discussion around recent literature, elaborate on how statistical model parsimony is handled, have a more moderated tone when describing the linkage between tapered observations with modelled results, and explore more TGR correlations. A more detailed list of my thoughts follows below.

Response: We thank the reviewer for reviewing our manuscript and for the constructive and encouraging comments. We have considered all of the suggestions. Below we provide detailed responses to each point raised by the reviewer.

1. I recently published a paper that covers a very similar topic [Schultz et al., 2025].

Statistical methods were used to infer tapers in the GR-MFD, albeit with a different approach. However, that paper interpreted the cause differently: suggesting it was related to the growth of a finite fracture for bound cases, while unbound were injection into larger pre-existing faults. I feel it would be appropriate for the authors to discuss the overlap/differences with this work. I have some thoughts that I’ll break into pieces.

Schultz, R., Lanza, F., Dyer, B., Karvounis, D., Fiori, R., Shi, P., ... & Wiemer, S. (2025). The bound growth of induced earthquakes could de-risk hydraulic fracturing. *Communications Earth & Environment*, <https://doi.org/10.1038/s43247-025-02881-2>.

a. To be clear, I’m not suggesting that my paper undermines the novelty of this work. To the contrary, I think that the author’s work nicely complements my study. As well, reproducing results is an important part of the scientific method.

b. The parts about earthquake complexity having a relationship with TGR cases (Line 117) corroborates this alternative interpretation.

c. We also have a pre-print online for a follow-up work [Schultz et al., 2026]. This paper goes into more detail on the interpretation: it uses other geophysical information to show that new fracture stimulation cases also tend to be bound sequences, while injection directly into a fault is typically unbound.

Schultz, R., Villiger, L., Gischig, V., & Wiemer, S. (2026). Interpreting the cause of bound earthquakes at underground injection experiments. *EGUsphere* [preprint], 2026, 1-49., <https://doi.org/10.5194/egusphere-2025-5806>.

d. Much of the methods and application described in the final section (Line 172+) are also similar to the other paper.

Response:

We thank the reviewer for bringing this very relevant reference to our attention.

The interpretation proposed in Schultz et al. (2025) is broadly consistent with the mechanisms discussed in our manuscript. We show that two different factors may limit the growth of induced earthquakes depending on the setting: (1) heterogeneous fault loading near injection sites, and (2)

structural control related to whether seismicity occurs within a finite fracture network or involves the reactivation of a larger pre-existing fault. The latter mechanism is consistent with the mechanism suggested by Schultz et al. (2025), as now acknowledged in our manuscript. In our study, this possibility is supported by the observation that TGR cases tend to exhibit higher fractal dimensions in the spatial distribution of seismicity, as noted by the reviewer. The role of heterogeneous loading is supported by our numerical simulations, which show that near-field injection scenarios can produce heterogeneous pore-pressure distributions along faults and may limit rupture growth. Consistent with the interpretation of heterogeneous fault loading, the observed seismicity in the TGR cases tends to occupy relatively small seismic cloud volumes. We have mentioned both mechanisms in the Abstract and main text. Please see Lines 13–15, 172–189, and 232–236.

Following the reviewer’s comment, we have now added citations to Schultz et al. (2025) in the Introduction, when discussing the spatial complexity of TGR cases, and also in the final section discussing (real-time monitoring). Please see Lines 45, 131–134, and 193–194. We also thank the reviewer for pointing to the new contribution. We are pleased to learn that their findings seem consistent with our analysis.

2. A general critique of the paper is that the linkage between tapered GR-MFDs and the earthquake sequence simulations is only suggestive, rather than truly indicative. Effectively, the heterogeneous loading of a fault is one way you could get a tapered GR-MFD, but not the only one. Not only that, but multiple different mechanisms could be at play. The tone of the paper should reflect this uncertainty (and most of the paper does accomplish this). However, in the abstract on Lines 15-16 or on Lines 213-215 should be revised with this in mind.

Response: We agree with the reviewer that the observed TGR behavior may arise from different mechanisms and may vary from site to site. To better reflect this uncertainty, we have revised the relevant sentences in the Abstract (Lines 13–17) and the main text (Lines 233–236) to avoid overly definitive wording.

3. As far as I can tell from manuscript text, the likelihood ratio test doesn’t include a penalty term for the difference in model complexity. I did investigate the original ‘methods’ paper, where this does seem to be addressed. It would be worthwhile for the authors to mention that this parsimony consideration is already accounted for in the maintext of this manuscript.

Response:

The likelihood-ratio test already accounts for the difference in model complexity through the χ^2 distribution of the statistic $2LR$, which represents twice the log-likelihood ratio between the GR and TGR models. The degrees of freedom correspond to the difference in the number of parameters between the two models. For example, when testing at the 5% significance level, the GR distribution would be rejected if $2LR > 3.84$.

We thank the reviewer for this comment, which made us realize that this point might not be sufficiently clear to readers. We have therefore clarified this in the Methods section (Lines 266–270) and in the main text (Lines 71–72).

4. The exploration of correlations with other characteristics is quite interesting.

a. How exactly are parameters normalized for comparison? For a hypothetical example regarding depth, if all of the TGR cases were from underground labs and the GR cases were from field-scale operations, then you could produce a ‘correlation’ that doesn’t have any meaning.

Response:

This is an important concern. To take into account potential scale effects in the original manuscript, we compared TGR and GR cases both (1) using the full dataset, including underground laboratory experiments (referred to as “small-scale experiments” in the manuscript) and (2) using only field-scale operations, so excluding the “small-scale experiments”. Please see Methods, Lines 371–374. This distinction is necessary because some parameters, such as depth, can depend strongly on operational scale, as the reviewer notes, whereas others, such as the fractal dimension of pairwise event distances, are expected to be largely scale-independent. In the original manuscript, values of all parameters for the complete dataset are shown in Supplementary Fig. 6 (now Supplementary Fig. 8), where circles outlined in black denote small-scale experiments. The cumulative distribution functions (CDFs) comparing GR and TGR sequences after excluding small-scale experiments are shown in Extended Data Fig. 3 (now Supplementary Fig. 9). We also generated CDFs including all sequences but did not include them in the manuscript for simplicity. In the original manuscript, Figs. 2a–e (including depth) already show indices calculated using only field-scale cases, thereby avoiding potential bias from laboratory experiments (as mentioned in the figure caption). In contrast, the fractal dimension analysis (Fig. 2f) included both small-scale experiments and field-scale operations because this parameter is not expected to depend on scale. Nevertheless, we verified that repeating the analysis using only field-scale sequences for the fractal dimension yields consistent results.

The reviewer’s comment made us realize that this distinction was not explained clearly enough. We also noted that two other parameters shown in Fig. 2d and 2e, the aspect ratio of the seismic cloud and the largest-event radius normalized by seismic cloud length, are likewise not expected to depend on scale, yet they were treated differently from the fractal-dimension analysis in the original figure.

To make the presentation more consistent, we revised Fig. 2 so that all parameters are now shown using only field-scale operations (i.e., the fractal-dimension analysis now also excludes small-scale experiments). In addition, we clarified in the main text that we evaluated potential scale dependence and verified that including small-scale experiments does not change the conclusions for Fig. 2d–f. Please see Lines 114–116, 134–137, and 713–714.

b. Does the correlation with depth suggest that many of these TGR cases are getting their taper from strata-bound faults? Said another way, is there a difference seen between sedimentary cases versus crystalline basement cases?

Response: The reviewer raises an interesting point. We examined this possibility in the original manuscript (see the fifth column of Supplementary Table 2). We did not observe a clear correlation between the occurrence of TGR behavior and whether the seismicity occurred in sedimentary formations or crystalline basement. We now highlight this point more explicitly in the revised manuscript (Lines 120–122).

c. The less oblate cases tending to be TGR feels backwards from expectation. Is this correct?

Response: This results is indeed not completely intuitive. Our interpretation is GR sequences are observed in more oblate seismicity clouds because they follow planar structures resembling a single dominant fault surface. By contrast the TGR rupture are confined within 3-D volumes. This interpretation is consistent with our observation that TGR sequences exhibit higher fractal dimensions of pairwise event distances, indicating greater geometric complexity of the activated fault structures. Such behavior is expected when seismicity occurs on a more heterogeneous or

poorly organized fault network, rather than on a well-defined planar fault system, as also discussed in Schultz et al. (2025). We have explained this in more detail in the revised manuscript (Lines 126–132).

d. The modelling results suggest that distance from injection should correlate with TGR observations: TGR cases should typically be close to injection, with GR cases being further away. Is this seen in the data? Could be something to give the heterogeneous loading interpretation more (or less) credibility.

Response: Yes. The modeling results are consistent with the observations. Because the induced sequences analyzed here are quite diverse, ranging from single-well injections to regional-scale operations, it is difficult to define a consistent distance between earthquakes and the injection wells across all cases. Instead, we use the seismic cloud volume as a proxy. Smaller seismic cloud volumes represent near-field scenario, whereas larger seismic cloud volumes reflect more spatially distributed, far-field activity. Consistent with the modeling results, TGR sequences tend to exhibit smaller seismic cloud volumes (Fig. 2), suggesting that near-field heterogeneous loading limits the growth of large earthquakes. We have mentioned this point in the manuscript (Lines 169–171):

“The difference resulting from near- and far-field injection is consistent with the observation that GR sequences often exhibit larger, deeper, and more diffuse seismic clouds (Figs. 2a–c).”

5. With Figure 1, it would be helpful if the cut-off for the 5% confidence threshold was clearer. For example, I can immediately tell that the 17 Newberry case passes 10%, but I’m not sure which cases pass the 5% threshold.

Response: We thank the reviewer for the suggestion to clarify which cases meet the 5% confidence threshold in Fig. 1. We now report the p_{LRT} value for each case directly in the legend of Fig. 1 (to the right of panel b). For example, the original legend entry “17-Newberry-2014” is now shown as “17-Newberry-2014 (0.08)”. We also state in the figure caption that the values in parentheses correspond to the p_{LRT} values and mention that the cases are sorted by the p_{LRT} values. Please see Lines 689–690.

6. The following page contains a list of minor suggestions.

a. Line 24: A recent paper reviewed how the risk of induced seismicity is handled and made recommendations for good practices and open research questions [Zhou et al., 2024]. The topics are relevant for the introduction, and it could be cited there. Of course, feel no obligation to do so. Zhou, W., Lanza, F., Grigoratos, I., Schultz, R., Cousse, J., Trutnevyte, E., ... & Wiemer, S. (2024). Managing induced seismicity risks from enhanced geothermal systems: A good practice guideline. *Reviews of Geophysics*, 62(4), e2024RG000849, <https://doi.org/10.1029/2024RG000849>.

Response: Thank you for the suggestion. We have added this reference to the Introduction (Line 24) and final section regarding real-time monitoring (Lines 193–194).

b. Lines 76-81: I’m not sure I understand this point. Why would shutting in the operation before a large event occurs influence the GR/TGR shape?

Response: Our intention was to test the case where operations are shut in after, not before, the occurrence of a relatively large event. Such operational responses can truncate the seismic sequence and introduce non-stationarity in the catalog. In principle, this could raise the concern that using such sequences may bias the estimation of the intrinsic magnitude–frequency distribution. To evaluate this possibility, we performed synthetic tests. The results show that such

termination does not produce TGR-like behavior and therefore does not bias our conclusions (current Supplementary Fig. 3). Previous studies have adopted a conservative approach to this issue; for example, van der Elst et al. (2016, JGR) analyzed only earthquakes occurring prior to the maximum-magnitude event to avoid potential non-stationarity. Our tests indicate that including the full catalog does not introduce a systematic bias in our analysis. We have revised the manuscript to clarify this point (Lines 81–87).

c. Supp refs #44 #47 have the wrong titles for the given dois.

Response: Thank you for pointing this out. We have corrected the titles accordingly. We also carefully checked the remaining references to ensure that the reference list does not contain any incorrect information.

d. Table S1: “Peace Rive” should be “Peace River”

Response: Thank you for pointing this out. Change made.

B. Response to Reviewer #2:

This study builds on methods developed in a previously published study (Li and Avouac, BSSA, 2025) to examine the magnitude-frequency distributions of induced earthquake sequences and determine whether they are better fit with a standard Gutenberg-Richter (GR) relationship or a tapered GR, and how this can evolve over time during each sequence. The results suggest that the size of induced earthquakes tends to be more limited than expected by the standard GR relationship, and numerical simulations suggest this is due to heterogeneous loading and fault network conditions. The conclusions seem well supported, the manuscript is generally well-written and easy to follow, and the topic is of great interest to both the scientific community and can have practical applications as well when it comes to injection operation management. I only have a few minor comments and suggestions that are aimed at adding a little more detail and/or clarity at certain points listed below.

Response: We thank the reviewer for the positive and thoughtful evaluation of our manuscript. We appreciate the reviewer's suggestions for improving the clarity of several aspects of the study. We have addressed all comments by adding clarifications and revising the manuscript where appropriate. Detailed responses to each point are provided below.

1. First, a general question. Perhaps I missed this in either the main manuscript or the Methods section, but why do figures such as Extended Data Fig. 1-2, and Extended Data Fig. 5 show confidence intervals for the GR results but not the TGR results?

Response:

We thank the reviewer for pointing this out. We acknowledge that the original manuscript did not clearly explain these choices, which may have caused confusion. We have now clarified this in the revised manuscript and figure captions. Detailed explanations are provided below.

For the examples shown in Fig. 1b and original Extended Data Fig. 1 (now Supplementary Fig. 2), confidence intervals are shown only for the GR distributions. The purpose of displaying these intervals is to illustrate that in many cases, the observed magnitude-frequency distributions deviate significantly from the GR model. This visual comparison supports the results of the likelihood-ratio test used to distinguish between GR and TGR models. Because the focus of these panels is to demonstrate the inconsistency between the observations and the GR model, showing the GR confidence intervals is sufficient for this purpose. We make sure this point is clear in the revised manuscript (Lines 76 to 79). Adding the confidence intervals for the TGR models could bring confusion.

For original Extended Data Fig. 2 (now Supplementary Fig. 6), we cannot show a simple shaded area as the confidence interval for the TGR cases as we do for the GR cases. The confidence interval shown for the GR distribution is derived from the theoretical uncertainty of the maximum magnitude under the GR model by Eq. (4). Importantly, this interval does not depend on catalog size once a b -value is assumed. For example, assuming $b = 1$ (as in the figure), Eq. (4) yields a universal 90% confidence interval of $[-0.48, 1.29]$ relative to the expected maximum magnitude a/b . We therefore display this interval as a reference shaded region. Different sequences have different estimated b -values, but we chose to show the interval only for $b = 1$ to avoid overcomplicating the figure. In contrast, the confidence interval of the TGR maximum magnitude is considerably more complex (see Supplementary Fig. 5). It depends on catalog size, b -value, and corner magnitude, and therefore varies among sequences and time windows. Displaying these intervals directly in the figure would make the visualization difficult to interpret. For this reason, we only state in the Methods section that results obtained using the 5th and 95th percentiles yield

qualitatively consistent results (Lines 334–336). To further clarify this point, we have added the following sentence to the caption of original Extended Data Fig. 2 (now Supplementary Fig. 6): “*For simplicity, we do not show the confidence interval for the predicted maximum magnitude because it depends on catalog size, b-value, and corner magnitude and therefore strongly varies across sequences and time windows.*”

Finally, in the original Extended Data Fig. 5 (now Supplementary Fig. 12) and Fig. 4, the two curves (originally shown in blue and red) represent the predicted maximum magnitudes (using either model) and the observed maximum magnitudes, respectively. The observed maximum magnitudes do not have associated confidence intervals, and therefore only one confidence interval is shown in the plots. Because blue and red are used elsewhere in the manuscript to represent the GR and TGR models, respectively, this may have caused confusion. To avoid this ambiguity, we have changed the colors in these figures to purple and brown in the revised manuscript.

2. L. 287: How large is “large n”?

Response: We thank the reviewer for pointing this out. In the revised manuscript we clarify that the approximation used in Eq. (4) only differs by the exact expression less than 0.01 when catalog size exceeds 100 (minimum catalog size considered in this study). Please see Lines 314–318.

3. L. 312-320: It would be helpful to add a little more detail to this paragraph, especially to clarify what some of the parameters are and how they are measured. For example, how is spatial fractal dimension measured? How is seismic front migration measured? And do the operational and geological parameters all come from the sources that are referred to in Supp. Table 1?

Response: Thank you for the suggestion. We have expanded this paragraph to clarify how these parameters are defined and measured. We have also clarified that the operational and geological parameters are compiled from the sources listed in Supplementary Table 1. Please see Lines 344–356, 361–362, and 375–383.

4. L. 321-331: It would be helpful to state that the results of the quantitative analysis can be found in Extended Data Figure 3.

Response: Thank you for the suggestion. Change made. Please see Lines 363–364.

5. L. 335-340: Likewise, please state where these Pearson correlation results can be seen.

Response: Thank you for the suggestion. Change made. Please see Lines 386.

6. Figure 2 caption (L. 654): Replace “versus” with “divided by”

Response: Change made. Please see Line 710.

7. Figure 4 caption (L. 691): In the sentence beginning “Decatur consistently follows the Gutenberg-Richter (GR) model...” shouldn’t this be the “tapered Gutenberg-Richter” model?

Response: Thank you for pointing this out. Yes, it should be “tapered Gutenberg–Richter” model. We have corrected it. Please see Lines 747–748.

8. Supplemental Text 1: The second paragraph is repeated.

Response: Thank you for pointing this out. We have deleted it.

9. Supplemental Fig. 6: Please define all the parameters that are shown in the various panels (as in Extended Data Fig. 3).

Response: Thank you for the suggestion. Change made. Please see caption of current Supplementary Fig. 8.

10. Supplemental Fig. 7: Why aren't the different sequences numbered as they are in Supp. Fig. 6? Also again please define what the parameters are that are shown in each panel.

Response: Thank you for the suggestion. Change made. Please see current Supplementary Fig. 10.

11. Supplemental Fig. 8: In the caption, it would probably be better to change the sentence beginning "The second to last row displays..." to "The second to last rows display..." (or "The, second, third, and last rows..." as in the following sentence), otherwise it sounds like it's only referring to the one row second from the bottom.

Response: We changed it to "The second, third, and last rows display...". Please see caption of current Supplementary Fig. 13.

12. Supplemental Table 1: In addition to WFD, the table also has "WWD" listed (e.g., Guthrie, Guy-Greenbrier). I assume this is the same as WFD (wastewater disposal = waste fluid disposal)?

Response: Thank you for pointing this out. Yes, they are the same. We changed all "WWD" to "WFD".

13. Supplemental Table 2: Please restate what the different operation abbreviations stand for. Also, how is "highly earthquake-active zone or not" defined?

Response: We have now redefined all abbreviations used in Supplementary Table 2. The classification of "highly earthquake-active zone" is qualitative and is based on descriptions in the cited literature, indicating whether abundant tectonic earthquakes were reported in the study area prior to injection operations. This explanation has been added as a table note in the revised manuscript.

C. Response to Reviewer #3:

In this paper, the authors present an analysis of observed magnitude-frequency distributions (MFD) in injection-induced seismic sequences. While they initially compiled over 100 cases worldwide, only 38 sequences were retained for detailed analysis due to the high quality of their data. Using statistical approaches, the authors evaluate whether each sequence follows the standard Gutenberg-Richter (GR) model or a Tapered Gutenberg-Richter (TGR) distribution. Traditionally, the GR model, which assumes a constant exponential decay in earthquake frequency with magnitude, is used to characterize seismicity. However, the authors observed that the standard model is inadequate in approximately half of the analyzed cases. Instead, a TGR distribution is preferred, as it accounts for a sharp downturn in the occurrence of larger events near a critical value known as the corner magnitude. This finding indicates that, at many sites, the physical conditions of injection effectively limit the potential magnitude of induced events. The authors conducted three-dimensional numerical simulations to investigate the physical mechanisms causing the tapering of the MFD. These simulations were performed on a planar fault, considering both homogeneous and self-affine heterogeneous properties, subjected to constant fluid injection. For these calculations, they utilized laboratory-derived rate-and-state friction laws to model the interaction between fault surfaces. In the simulations, the authors demonstrate that fault geometry and stress heterogeneity within the fluid-perturbed volume are what constrain the maximum size of earthquakes, rather than the spatial extent of the perturbation itself. Finally, the authors propose a practical strategy for real-time seismic risk management during injection operations. The Gutenberg-Richter (GR) model is initially assumed to define the maximum allowable injection volumes based on acceptable magnitude thresholds. As operations proceed, statistical tests, specifically the likelihood-ratio test developed by authors in previous work, are employed to assess whether the seismic sequence continues to follow the GR model or if a Tapered Gutenberg-Richter (TGR) distribution provides a superior fit. This distinction is crucial: a TGR classification indicates that the probability of large-magnitude events is significantly lower than the standard model predicts. This framework allows operators to adjust injection parameters, such as volume and pressure, to mitigate the hazard and prevent the triggering of damaging earthquakes.

The manuscript is clearly written, the methodology is thoroughly explained, the data sources are provided in the Supplementary Table 1, and the code used is properly referenced.

The methodological approach is innovative and compelling. It establishes a fundamental framework for artificial injection systems, providing a robust basis for evaluating seismic hazard before and during subsurface operations.

I recommend that the paper be accepted as it is.

Response: We appreciate the reviewer's thoughtful summary of the work and their positive assessment and feedback.

Response to Reviewer #1 (Ryan Schultz):

The revised manuscript of Li et al., “Large-magnitude events unlikely in induced earthquake sequences” is an article concerning the shape (and cause) of the statistical distribution of induced earthquake magnitudes.

In this revised version, the authors have suitably addressed all of my concerns – I am happy to recommend it for publication now.

There’s just one small typo I noticed, after re-reading: on Line 132, “Schutz” should be “Schultz”.

Response: We thank the reviewer for carefully reviewing our revised manuscript and for the positive evaluation. The typo has been corrected.

The manuscript of Li et al., “Large-magnitude events unlikely in induced earthquake sequences” is an article concerning the shape of the statistical distribution of induced earthquake magnitudes. The authors use statistical methods to identify cases of induced seismicity that deviate from unbound magnitudes. They perform numerical earthquake sequence simulations to suggest that this deviation can be related to heterogeneous loading of stress on a rough fault. Generally, the paper is well-written, logical, timely, and covers an interesting topic. I feel that the results of the paper could be interesting the readership of Nat. Comm. As a caveat, my expertise with earthquake sequence simulations is limited to just the implications.

That said, I feel there are additions/revisions that are needed to improve this paper before I would recommend it for publication. In particular, the manuscript could bolster some discussion around recent literature, elaborate on how statistical model parsimony is handled, have a more moderated tone when describing the linkage between tapered observations with modelled results, and explore more TGR correlations. A more detailed list of my thoughts follows below.

1. I recently published a paper that covers a very similar topic [Schultz et al., 2025]. Statistical methods were used to infer tapers in the GR-MFD, albeit with a different approach. However, that paper interpreted the cause differently: suggesting it was related to the growth of a finite fracture for bound cases, while unbound were injection into larger pre-existing faults. I feel it would be appropriate for the authors to discuss the overlap/differences with this work. I have some thoughts that I’ll break into pieces.

Schultz, R., Lanza, F., Dyer, B., Karvounis, D., Fiori, R., Shi, P., ... & Wiemer, S. (2025). The bound growth of induced earthquakes could de-risk hydraulic fracturing. *Communications Earth & Environment*, <https://doi.org/10.1038/s43247-025-02881-2>.

- a. To be clear, I’m not suggesting that my paper undermines the novelty of this work. To the contrary, I think that the author’s work nicely complements my study. As well, reproducing results is an important part of the scientific method.
- b. The parts about earthquake complexity having a relationship with TGR cases (Line 117) corroborates this alternative interpretation.
- c. We also have a pre-print online for a follow-up work [Schultz et al., 2026]. This paper goes into more detail on the interpretation: it uses other geophysical

information to show that new fracture stimulation cases also tend to be bound sequences, while injection directly into a fault is typically unbound.

Schultz, R., Villiger, L., Gischig, V., & Wiemer, S. (2026). Interpreting the cause of bound earthquakes at underground injection experiments. *EGUsphere [preprint]*, 2026, 1-49., <https://doi.org/10.5194/egusphere-2025-5806>.

- d. Much of the methods and application described in the final section (Line 172+) are also similar to the other paper.
2. A general critique of the paper is that the linkage between tapered GR-MFDs and the earthquake sequence simulations is only suggestive, rather than truly indicative. Effectively, the heterogeneous loading of a fault is one way you could get a tapered GR-MFD, but not the only one. Not only that, but multiple different mechanisms could be at play. The tone of the paper should reflect this uncertainty (and most of the paper does accomplish this). However, in the abstract on Lines 15-16 or on Lines 213-215 should be revised with this in mind.
 3. As far as I can tell from manuscript text, the likelihood ratio test doesn't include a penalty term for the difference in model complexity. I did investigate the original 'methods' paper, where this does seem to be addressed. It would be worthwhile for the authors to mention that this parsimony consideration is already accounted for in the maintext of this manuscript.
 4. The exploration of correlations with other characteristics is quite interesting.
 - a. How exactly are parameters normalized for comparison? For a hypothetical example regarding depth, if all of the TGR cases were from underground labs and the GR cases were from field-scale operations, then you could produce a 'correlation' that doesn't have any meaning.
 - b. Does the correlation with depth suggest that many of these TGR cases are getting their taper from strata-bound faults? Said another way, is there a difference seen between sedimentary cases versus crystalline basement cases?
 - c. The less oblate cases tending to be TGR feels backwards from expectation. Is this correct?

- d. The modelling results suggest that distance from injection should correlate with TGR observations: TGR cases should typically be close to injection, with GR cases being further away. Is this seen in the data? Could be something to give the heterogeneous loading interpretation more (or less) credibility.
5. With Figure 1, it would be helpful if the cut-off for the 5% confidence threshold was clearer. For example, I can immediately tell that the 17 Newberry case passes 10%, but I'm not sure which cases pass the 5% threshold.
6. The following page contains a list of minor suggestions.

Thanks,

-Ryan

Line 24: A recent paper reviewed how the risk of induced seismicity is handled and made recommendations for good practices and open research questions [Zhou et al., 2024]. The topics are relevant for the introduction, and it could be cited there. Of course, feel no obligation to do so. Zhou, W., Lanza, F., Grigoratos, I., Schultz, R., Cousse, J., Trutnevyte, E., ... & Wiemer, S. (2024). Managing induced seismicity risks from enhanced geothermal systems: A good practice guideline. *Reviews of Geophysics*, 62(4), e2024RG000849, <https://doi.org/10.1029/2024RG000849>.

Lines 76-81: I'm not sure I understand this point. Why would shutting in the operation before a large event occurs influence the GR/TGR shape?

Supp refs #44 #47 have the wrong titles for the given dois.

Table S1: "Peace Rive" should be "Peace River"